# UniBoost: Boost Zero-shot Vision-Language Tasks via Multitask Fine-tuning with Unsupervised Unimodal pre-training

## Abstract

Large-scale joint training of multimodal models, e.g., CLIP, have demonstrated great performance in many vision-language tasks. However, pre-training with image-text pairs limits itself to cover a wide range of unimodal data, where noise can also be introduced as misaligned pairs during pre-processing. Conversely, unsupervised training of unimodal models on text or image data alone can achieve broader coverage of diverse real-world data. This motivates us to propose a method based on unsupervised pre-trained unimodal models to enhance the zero-shot performance for vision-language tasks. Overall, our method is a multitask fine-tuning framework initialized from separate unsupervised pre-trained vision and language encoders, which allows the model to benefit from both the unsupervised pre-training and a variety of supervised data. Experiments show that our method outperforms state-of-the-art CLIP-based models by 6.5% (52.3% $\to$ 58.8%) on PASCAL-$5^i$ and 6.2% (27.2% $\to$ 33.4%) on COCO-$20^i$ under zero-shot language-guided semantic segmentation setting respectively. By learning representations of both modalities, unimodal pre-training offers strong generalization ability, while multitask fine-tuning shares knowledge across tasks and enhances domain adaptation, resulting in better performance especially for zero-shot vision-language tasks.

## 1 Introduction

Vision-language tasks have attracted much attention, producing excellent results by capitalizing on language-supervised pre-trained models such as CLIP Radford et al. (2021). Motivated by the strong generalization ability of CLIP, extensive effort has been made to transfer the knowledge in image-text pair pre-trained models to downstream tasks, e.g., open-vocabulary detection Gu et al. (2022b); Zang et al. (2022a) and segmentation Xu et al. (2022); Ding et al. (2022); Li et al. (2022a).

However there are two main limitations of such image-text pair models. One is the limiting size of image-text pairs that restricts their ability to cover a large distribution of real-world data. The other limitation is that they require a significant amount of pre-processing, where errors can be introduced in aligning image and text data which hinder their accuracy. In contrast, unimodal models that are trained through unsupervised techniques on text or image data alone can achieve a much broader coverage of the real-world distribution of data, see Figure 1. This is because these models are not constrained by the requirement that image and text data to be present simultaneously for pair-up. Unimodal models are often trained on larger datasets with much less pre-processing and no pair-ups, and thus causing no such errors, making them more flexible in handling different types of data.

On the other hand, current trend of vision-language tasks adopts pretrain-then-finetune paradigm, where the model is first pre-trained from scratch on large-scale image-text pairs, and then fine-tuned on specific task. To bridge the gap between the domain of pre-traing datasets and the specific domain of downstream tasks, another intermediate fine-tuning step is usually introduced before the specific task fine-tuning. Intermediate fine-tuning is a method of fine-tuning a pre-trained model using an intermediate task, which includes domain adaption or task adaptation. Fine-tuning on a similar domain or similar task with relatively large-scale supervised data makes the pre-trained model well adapted to the specific target domain or task. In this paper, we extend the intermediate fine-tuning to multitask fine-tuning so as to benefit from the shared knowledge across multiple tasks and datasets while keeping the adaptation ability for different specific tasks.

In general, our method, called UniBoost, is a multitask fine-tuning framework based on the unsupervised pre-trained vision and language models. Unimodal pre-trained models can make use of all available image and text data rather than limited to image-text pairs. Despite the strong generalization ability of isolate vision or language models, downstream vision-language tasks require the alignment or fusion of image and text space. Therefore, we introduce multitask fine-tuning to achieve the multimodal alignment or fusion. Instead of aligning or fusing the two spaces on an extensive large-scale image-text pairs such as LAION used in Li et al. (2023), we directly align or fuse the vision and language space on a variety of tasks with supervision. Fine-tuning on multiple tasks with supervised data allows our framework to achieve both general multimodal alignment or fusion across diverse data, with the advantage of intermediate fine-tuning for the adaptability to downstream tasks.

From our experiments, we find that our framework, which takes the advantages of unsupervised unimodal pre-training as well as multitask fine-tuning, boosts the zero-shot performance for vision-language tasks. In addition, our experiments show that unsupervised unimodal pre-training is surprisingly more effective than supervised pre-training or image-text pair supervised pre-training under multitask fine-tuning. Specifically, zero-shot language-guided segmentation based on two unimodal pre-trained weights, namely, MAE He et al. (2022) and T5 Raffel et al. (2020), outperforms the models based on CLIP model by more than 5% on mIoU under similar model capacity.

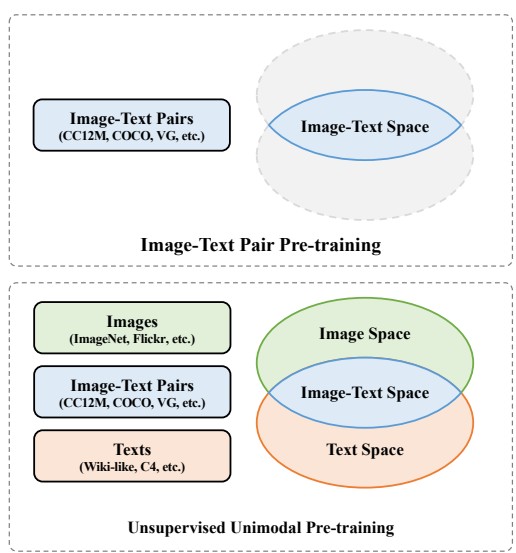

Figure 1: The solution space of conventional vision-language pre-training is restricted to image-text pairs while unsupervised unimodal models can be trained on not only image-text pairs, but also image or text data alone, i.e., a broader range of data distribution. Our UniBoost, a multitask fine-tuning framework based on unsupervised unimodal supervised models, benefits from the general and robust representations of unsupervised unimodal models as well as diverse supervised data and thus boost zero-shot vision-language tasks.

Compared to unified models Wang et al. (2023); Lu et al. (2023), our method is based on separate pre-trained image and language models instead of learning the representation from scratch, making our method easily benefit external latest unimodal models. Compared to the models Li et al. (2023); Zou et al. (2023) with separate vision and language encoders, our method are directly fine-tuned on diverse tasks instead of learning from image-text pairs, making our method easily adapt to specific downstream task. Overall, our method encompasses the following contributions. First, UniBoost is built on separate unsupervised pre-trained unimodal encoders, which can help leverage the strong generalization ability from the powerful language or vision models flexibly. Second, UniBoost can achieve multimodal alignment or fusion, as well as domain and task adaptation through multitask fine-tuning with diverse supervised data simultaneously, which can help regularize the model, prevent overfitting and share common knowledge, as the model is exposed to a wider range of data and tasks during multitask fine-tuning.

## 2 RELATED WORK

### 2.1 VISION-LANGUAGE MODELS (VLMS)

Vision-Language Models (VLMs) aim to bridge the gap between visual and textual modalities. VLM has been intensively investigated recently on various multimodal tasks, e.g., visual question answering (VQA) Peng et al. (2020); Hu et al. (2019), image/video captioning Pan et al. (2004); Kulkarni et al. (2013); Li et al. (2019); Zhang & Peng (2019), visual grounding Mao et al. (2016); Yu et al. (2016); Liu et al. (2017), referring segmentation Ye et al. (2019); Feng et al. (2021); Ding et al. (2021) and text-to-image generation Reed et al. (2016); Gu et al. (2022a). Recently, vision tasks such as instance segmentation Li et al. (2022a); Ma et al. (2022) and object detection Gu et al. (2022b); Du

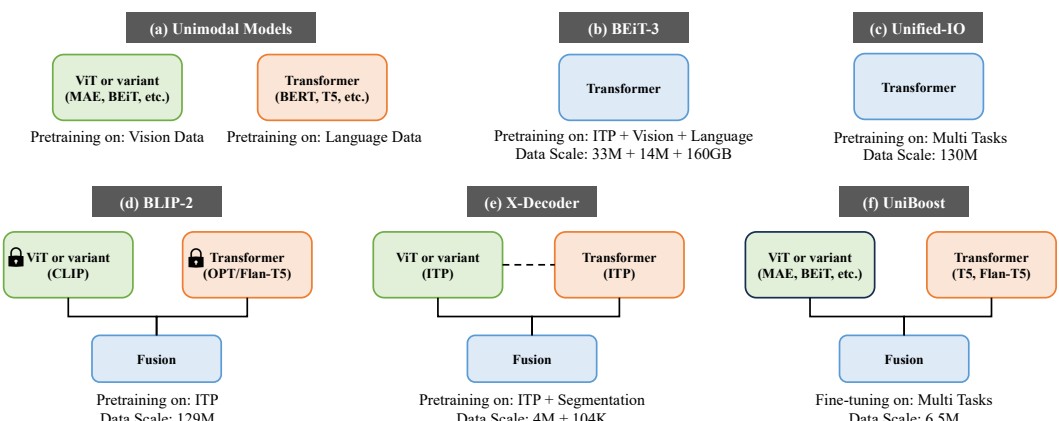

Figure 2: Comparisons among existing popular methods Wang et al. (2023); Lu et al. (2023); Li et al. (2023); Zou et al. (2023) and our UniBoost. ITP is short for image-text pairs.

et al. (2022) introduce text embeddings into the model designs to boost performance or implement open-vocabulary pixel or object classification. Specifically, VLM pre-training Radford et al. (2021); Jia et al. (2021); Yu et al. (2022); Wang et al. (2022a); Alayrac et al. (2022); Li et al. (2022e); Bao et al. (2022b); Wang et al. (2023) has provided strong and effective foundation models for these multimodal applications. The VLMs are pre-trained to learn rich vision-language correspondences from large-scale informative image-text pairs supervised by certain vision-language objectives He et al. (2020); Devlin et al. (2019); Yang et al. (2022); He et al. (2022); Singh et al. (2022); Li et al. (2022c). The pre-trained high-capacity VLMs can perform zero-shot predictions on downstream applications even without fine-tuning by matching the feature embeddings of images and texts directly. However, despite their success, existing VLMs are trained on image-text pairs of the intersection of image and text data. By contrast, we propose to explore the union of image and text data for VLMs, which is fundamentally and theoretically much larger than the image-text pairs data. Different from Bao et al. (2022b); Wang et al. (2023) utilizing a multi-way transformer to pre-train on image-text pair data and image or text data alone from scratch, we directly transfer the knowledge from well pre-trained unimodal models and learn the multimodal alignment or fusion through intermediate fine-tuning on in-domain tasks, which is more efficient and flexible.

## 2.2 Unsupervised Pre-training (UPT)

Unsupervised Pre-training fuels the progress of VLM foundation models by enabling effective usage of massive internet data for model pre-training. It significantly reduces the dependency on data labeling, where the encoder networks are trained with self-supervision by specific pretext tasks. Early UPT works design special pretext tasks and train the model to predict the corresponding answers, e.g., context prediction Doersch et al. (2015), inpainting Pathak et al. (2016), colorization Zhang et al. (2016), jigsaw Noroozi & Favaro (2016), visual primitives counting Noroozi et al. (2017), and rotation prediction Komodakis & Gidaris (2018). The contrastive learning based UPT He et al. (2020); Chen et al. (2020); Jia et al. (2021); Yang et al. (2022); Oord et al. (2018); Khosla et al. (2020) trains the model by learning the prior knowledge distribution of the data itself by making similar instances closer and dissimilar instances farther apart in the feature space. Recently, the autoencoder-based masked prediction methods have demonstrated great effectiveness on the large-scale foundation model pre-training. Such methods He et al. (2022); Xie et al. (2022); Li et al. (2022d); Chen et al. (2022); Singh et al. (2022); Bao et al. (2022a); Luo et al. (2022) train the model to recover the masked image patches from a corrupted input image. Our work is built on unsupervised unimodal pre-trained models, to fully leverage their generalization ability for boosting VLM tasks.

## 2.3 multitask Fine-tuning (MFT)

Task Fine-tuning (TFT) is an essential step to transfer the high-capacity model from the pre-training domain to specific downstream tasks and applications. TFT techniques involves prompt tuning Zhou et al. (2022b;a); Ma et al. (2023); Derakhshani et al. (2022); Jia et al. (2022); Bahng et al. (2022); Zang et al. (2022b); Shen et al. (2022); Khattak et al. (2022); Xing et al. (2022), feature adaptation Gao et al. (2021); Zhang et al. (2021b); Pantazis et al. (2022); Houlsby et al. (2019), direct fine-tuning Wortsman

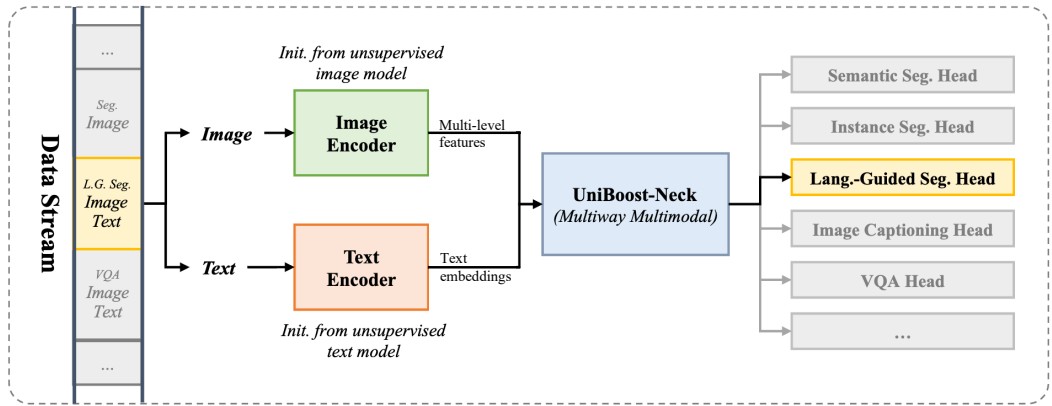

Figure 3: **UniBoost** is a multitask fine-tuning framework based on unsupervised unimodal encoders. The input can be image, text or image-text pair depending on the target task during multitask fine-tuning. The UniBoost-Neck is a multiway multimodal module as shown in Figure 4
.

et al. (2022), and architecture modification Zhou et al. (2021). multitask fine-tuning (MTF) extends the fine-tuning from single task to multitasks, so as to leverage the shared information of multiple related tasks Shen et al. (2022); Caruana (1997); Dong et al. (2019); Sun et al. (2019; 2020); Ren & Lee (2018) and improve the generalization of all the tasks. MTF can optimize the model on multiple disjoint datasets, erasing the demanding for the expensive exhaustive labeling and simultaneously maintaining high performance on all tasks. Despite its benefits such as enhanced data efficiency and decreased over-fitting, MFT still poses challenges regarding negative transfer and imbalanced dataset sizes. To address these challenges, a common solution is to weight per-task losses or combine per-task gradients into a joint update Chen et al. (2018); Kendall et al. (2018); Liu et al. (2021); Sener & Koltun (2018). However, these works require the MFT model to have at least one forward propagation on each task so that they can manipulate the overall losses or gradients, but this requirement cannot be easily satisfied, making these methods not directly applicable. Another solution is task sampling-based MTL (TS-MTL) in which only one task along with its data point is sampled per iteration. Task-sampling strategies Hu & Singh (2021); Jean et al. (2019); Lu et al. (2020) aims to balance different tasks, avoiding the over-fitting on data-poor tasks or catastrophic forgetting on data-rich tasks. However, it is found that TS-MTL often underperforms single ask trained models; it is thus typically followed by an additional per-task fine-tuning step.

## 3 UNIBOOST

The pretrain-then-finetune paradigm is very effective which holds the holy grail for modern vision and vision-language tasks. Recently, intermediate fine-tuning has been inserted before fine-tuning the pre-trained models on the target task for preventing overfitting and enhancing adaptation. Inspired by this, our **UniBoost**, a multitask fine-tuning framework as shown in Figure 3 based on the unsupervised pre-trained vision and language models that can not only fully leverage the generalization ability of unsupervised unimodal models, but also make use of multitask supervised data to optimize the alignment of image and text embedding space, thereby improving the ability of transferring knowledge to downstream tasks. Note that, the vision and language models we use are already pre-trained on large-scale image or text data alone, therefore, our method does not require a pre-training step. We directly learn the multimodal alignment or fusion through multitask fine-tuning.

### 3.1 MODEL ARCHITECTURE

Our method is a multitask fine-tuning framework based on separate image and text encoders for extracting image and text embeddings respectively. Considering the framework will be trained on multiple tasks which require different levels of information, we take multi-layer image features from the image encoder as well as text embeddings (if any) and send them to the neck. The neck works bridges the embeddings and the task head, where image and text embeddings are aligned or fused, or the multi-level information are aggregated. We elaborate the details in the following.

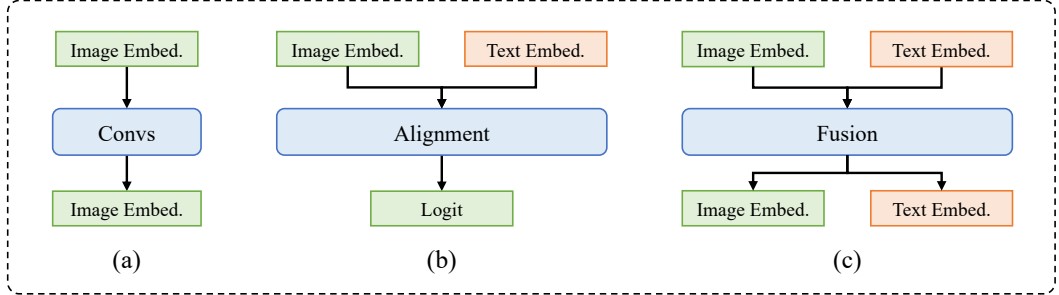

Figure 4: **UniBoost-Neck** is a multiway multimodal module which supports different types of vision tasks or vision-language tasks: (a) vision tasks which require image embedding only; (b) vision-language tasks which require aligned image and text embeddings, (c) vision-language tasks which require deep fusion of image and text embeddings.

The neck in UniBoost is a multiway multimodal module supporting different vision tasks or vision-language tasks: (a) vision tasks which require image embedding only, (b) vision-language tasks which require aligned image and text embeddings, (c) image-grounded text generation tasks which require deep fusion of image and text embeddings. Afterward, the unimodal or multimodal representations are sent to task-specific heads to produce predictions and compute losses. For clarity, let the image and text embedding extracted from the unimodal encoders are respectively denoted as $E_i \in \mathbb{R}^{H \times W \times C_i}$ and $E_t \in \mathbb{R}^{C_t}$. We use $\Omega$ to represent the trainable parameters used in the UniBoost-Neck.

**Vision tasks.** Vision tasks such as detection or segmentation usually rely on pyramid features to provide multi-level information. In this work flow, the neck inherits the conventional feature pyramid network design, and produces output image embedding $\tilde{E}_i = \Omega(E_i)$. Specifically, the neck takes multi-level features as input, append the pyramid spatial pooled features for enlarging receptive fields, build a top-down path for propagating high-level information progressively, and apply a conv-bn-relu block for producing features. Different task heads follow afterwards. For example, in detection, a fast-rcnn head is applied; in classification, a global average pooling layer is applied.

**Image-text alignment tasks.** Many conventional detection or segmentation methods are extended from close set to open set through replacing the fixed-length classification layer with class embeddings. Specifically, in open-set recognition tasks, we feed the $N$ queried class names into the text encoder to extract embeddings $\mathbf{E}_t \in \mathbb{R}^{N \times C_t}$ for all classes. For the text encoder which outputs separate embedding for each word instead of global embedding, we take the mean value of all the word embeddings if there are multiple words describing a class such as "dining table". The image embedding is enhanced in the same way as that in vision tasks. After obtaining the image and class embeddings, a projection layer is employed separately to obtain $E_i \in \mathbb{R}^{H \times W \times C}$ and $\mathbf{E}_t \in \mathbb{R}^{N \times C}$. Then we take the dot product of the projected image and text embeddings for producing the logit $\mathcal{S} = E_i \cdot \mathbf{E}_t^{\mathsf{T}}$. We treat this logit as the classification result instead of learning a classification layer.

**Image-text fusion tasks.** Image-text fusion tasks such as visual question answering require the deep interaction between image embedding and text embedding. To achieve this, we instantiate $\Omega$ with a transformer network on the two embeddings formulated as,

$$[\tilde{E}_i, \tilde{E}_t] = \Omega([E_i, E_t]) \tag{1}$$

Similarly, we project the image or text embedding into the same dimension first. Then we directly flatten the image embedding along the height and width dimension and concatenate the flattened image embedding with text embedding, which are then passed into a transformer network. We treat both VQA and image captioning as conditional generation tasks. Specifically, we employ a special self-attention mask which allows visual tokens to attend to each other in both directions among all image tokens while textual tokens can attend to image tokens and their leftward text tokens. During training, the model is trained to predict next token based on the visual context and available textual context with cross entropy loss. We use the prompt "Question: {}, Answer: " and "a photo of " as the text for VQA and captioning task respectively. During inference, we generate the text tokens one by one in auto-regressive manner. In implementation, we adopt the multimodal transformer Wang et al. (2023) which consists of self-attention layers and 3 types of FFNs, i.e. V-FFN, L-FFN and VL-FFN expert in processing different modalities. In each transformer block, the visual and textual tokens

Table 1: Comparisons for language-guided semantic segmentation on PASCAL-$5^i$.

| Method | Image Encoder | Text Encoder | Shot | $5^0$ | $5^1$ | $5^2$ | $5^3$ | mean | FB-IoU |
|---|---|---|---|---|---|---|---|---|---|
| PFENet Tian et al. (2022) | | - | 1-shot | 60.5 | 69.4 | 54.4 | 55.9 | 60.1 | 72.9 |
| RePRI Boudiaf et al. (2021) | RN101 | - | 1-shot | 59.6 | 68.6 | **62.2** | 47.2 | 59.4 | - |
| HSNet Min et al. (2021) | | - | 1-shot | **67.3** | **72.3** | 62.0 | **63.1** | **66.2** | **77.6** |
| SPNet Xian et al. (2019) | RN101 | - | zero-shot | 23.8 | 17.0 | 14.1 | 18.3 | 18.3 | 44.3 |
| ZS3Net Bucher et al. (2019) | RN101 | - | zero-shot | 40.8 | 39.4 | 39.3 | 33.6 | 38.3 | 57.7 |
| LSeg Li et al. (2022a) | ViT-L | CLIP-B | zero-shot | 61.3 | 63.6 | 43.1 | 41.0 | 52.3 | 67.0 |
| UniBoost (Ours) | MAE-L | CLIP-B | zero-shot | 67.3 | 65.1 | 46.7 | 47.3 | 56.6 | 69.4 |
| UniBoost (Ours) | MAE-L | T5-S | zero-shot | **68.7** | **67.1** | **49.0** | **50.4** | **58.8** | **70.8** |

will first pass through a shared self-attention layer and separate V-FFN as well as L-FFN respectively. Afterwards, another shared self-attention layer and a VL-FFN expert are employed.

## 3.2 MULTITASK FINE-TUNING

Existing vision-language paradigms tend to learn representations for image and text, as well as their alignment (or fusion) jointly. However, such joint learning restricts the embedding space to the intersection of images and texts. Differently, we disentangle the representation and alignment (or fusion) learning. In UniBoost, since the image and text encoders are already initialized from well pre-trained unsupervised unimodal models, with effective contrastive learning or masked modeling techniques on large-scale data, the extracted image and text embeddings are generalized and robust enough for transfer learning. Therefore, we only need to align (or fuse) the isolated image and text embedding space through multitask in-domain fine-tuning at a relatively low cost.

**Data and Multitasking.** There are numerous valuable public datasets with annotations which can contribute to the vision-language learning. For benefiting from them, UniBoost is trained on a variety of tasks, involving high-level tasks such as classification, detection, pixel-level tasks such as semantic segmentation, vision-language tasks such as VQA, image captioning. In total, the multitask fine-tuning is conducted on 12 tasks consisting of 6.5M images. Some datasets are of limited size which only comprises thousands of images. Before training, we augment those datasets through cropping and flipping for enriching the samples to alleviate the data imbalance issue. Within each batch, the data points are all sampled from the same task. During iterations, we follow a sampling strategy in which each task will be sampled at once within a number of iterations.

**Experiment Setup.** In UniBoost, we initialize the image encoder and text encoder from well pre-trained models, such as MAE He et al. (2022), BEiT Bao et al. (2022a) for the image encoder, and T5 Raffel et al. (2020) for the text encoder. All image encoders take $16 \times 16$ patch size and the input resolution is set to a moderate value $480 \times 480$. We train UniBoost for 1M iterations with a batch size of 64 on 8 cards of NVIDIA Tesla V100 with 32GB. We use the AdamW for optimization and a cosine learning rate schedule with a peak learning rate of 1e-4 and a linear warmup of 5K steps. Note that the learning rate for encoders is multiplied by a ratio of 0.1 to make the shared parameters across all tasks updated at a slower rate for stable convergence. The weight decay is 0.01.

## 4 EXPERIMENTS

We extensively evaluate UniBoost on popular vision-language benchmarks including language-guided semantic segmentation, language-guided object detection and instance segmentation, visual question answering, and image captioning.

### 4.1 LANGUAGE-GUIDED SEMANTIC SEGMENTATION

Language-guided semantic segmentation is a multimodal task using language information for pixel classification. This task replaces the convectional classification layer with class text embeddings for zero-shot segmentation. We evaluate our UniBoost under zero-shot setting following method Li et al. (2022a) and supervised setting following method Rao et al. (2022). The results for zero-shot setting are shown in Table 1 and Table 2. Table 3 shows the results under supervised setting.

Table 2: Comparisons for language-guided semantic segmentation on COCO-$20^i$.

| Method | Image Encoder | Text Encoder | Shot | $20^0$ | $20^1$ | $20^2$ | $20^3$ | mean | FB-IoU |
|---|---|---|---|---|---|---|---|---|---|
| PFENet Tian et al. (2022) | RN101 | - | 1-shot | 36.8 | 41.8 | 38.7 | 36.7 | 38.5 | 63.0 |
| HSNet Min et al. (2021) | | - | 1-shot | **37.2** | **44.1** | **42.4** | **41.3** | **41.2** | **69.1** |
| ZS3Net Xu et al. (2022) | RN101 | - | zero-shot | 18.8 | 20.1 | 24.8 | 20.5 | 21.1 | 55.1 |
| LSeg Li et al. (2022a) | ViT-L | CLIP-B | zero-shot | 28.1 | 27.5 | 30.0 | 23.2 | 27.2 | 59.9 |
| UniBoost (Ours) | MAE-L | CLIP-B | zero-shot | 30.4 | 31.8 | 35.7 | 33.5 | 32.9 | 61.9 |
| UniBoost (Ours) | MAE-L | T5-S | zero-shot | **31.0** | **33.2** | **35.9** | **33.6** | **33.4** | **62.3** |

Table 3: Comparisons for language-guided semantic segmentation on ADE20K.

| Method | Image Encoder | Text Encoder | mIoU | pixAcc |
|---|---|---|---|---|
| SETR Zheng et al. (2021) | ViT-B | - | 46.2 | - |
| Semantic FPN Kirillov et al. (2019) | ViT-B | - | 49.1 | - |
| BEiT3 Wang et al. (2023) + FPN | BEiT3-B | - | 48.4 | 83.2 |
| DenseCLIP Rao et al. (2022) | CLIP-ViT-B | CLIP-B | 50.6 | - |
| UniBoost (Ours) | BEiT-B | T5-B | **52.9** | **84.6** |

**Dataset.** Under the zero-shot setting, we conduct experiments on PASCAL-$5^i$ and COCO-$20^i$, which are popular few-shot segmentation datasets and are usually used in zero-shot segmentation methods to evaluate the generalization ability from seen classes to unseen classes. Specifically, PASCAL-$5^i$ split 20 classes into 4 folds, with each fold denoted as PASCAL-$5^i$, $i \in 1, 2, 3, 4$. In each fold, 5 classes with the responding mask annotations are taken as the novel set in evaluation while the others form the base set used in training. Similarly, COCO-$20^i$ also have 4 folds of 20 classes each. Under the supervised setting, we train and evaluate UniBoost on ADE20K.

**Experiment Setup.** For the zero-shot setting, UniBoost is fine-tuned on seen classes in each fold for 15 epochs with a batch size of 8. Augmentation includes random resized cropping, horizontal flipping, and color jittering. We use the SGD with a momentum of 0.9 for optimization following Li et al. (2022a). We use a linear learning rate schedule with a base learning rate of $5e^{-5}$. For the supervised setting, we fine-tune UniBoost for 80K steps with a batch size of 32. AdamW and a poly learning rate with the base learning rate as $1e^{-4}$ and 1500 warmup steps are used for optimization.

**Results.** In Table 1, we show the results of UniBoost with different pre-trained image and text encoders to validate the substantial improvement of unsupervised pre-training over supervised pre-training or image-text pair supervised pre-training. We find that models with unsupervised pre-trained weights consistently outperform models with supervised pre-trained weights or image-text pair supervised pre-trained weights. Specifically, based on the supervised baseline, if we replace the image encoder of ViT-L Dosovitskiy et al. (2021) pre-trained on ImageNet-1K with an unsupervised pre-trained image encoder, i.e., MAE-L He et al. (2022), the performance is improved to 56.6%. Then, if we further apply T5 Raffel et al. (2020) as the pre-trained text encoder, the performance is boosted to 58.8%. Note that our zero-shot results even out-perform the 1-shot results by HSNet Min et al. (2021) in $5^0$. Table 2 shows the results on COCO-$20^i$ dataset. With UniBoost, the performance is boosted from 27.2% to 33.4%. These experiments demonstrate the effectiveness of UniBoost on aligning unsupervised pre-trained unimodalities for better performance on zero-shot tasks.

Similar conclusion can be drawn for fully-supervised experiments in Table 3, where we replace the CLIP-based pre-trained weights by the unsupervised pre-trained weights on more data for both image encoder and text encoder, improving the performance from 50.6% to 52.9% mIoU on ADE20K under fully supervised setting. This also demonstrates the effectiveness of unsupervised pre-training compared to image-text pair supervised pre-training.

## 4.2 LANGUAGE-GUIDED OBJECT DETECTION AND INSTANCE SEGMENTATION

Similar to language-guided semantic segmentation, language-guided object detection or instance segmentation methods utilize class text embeddings as the classification weight for the detected

Table 4: Comparisons for language-guided object detection and instance segmentation on COCO. DenseCLIP with CLIP-RN101 is trained and evaluated with an input size of $1333 \times 800$ while other transformer-based models use an input size with maximum side of 800 due to memory limitation.

| Method | Image Encoder | Text Encoder | $AP^b$ | $AP^b_{50}$ | $AP^b_{75}$ | $AP^m$ | $AP^m_{50}$ | $AP^m_{75}$ |
|---|---|---|---|---|---|---|---|---|
| DenseCLIP Rao et al. (2022) | CLIP-RN101 | CLIP-B | 42.6 | 65.1 | 46.5 | 39.6 | 62.4 | 42.4 |
| DenseCLIP Rao et al. (2022) | CLIP-ViT-B | CLIP-B | 41.3 | 64.1 | 44.5 | 37.8 | 60.7 | 39.8 |
| UniBoost (Ours) | BEiT-B | T5-B | **44.3** | **67.3** | **48.8** | **40.8** | **63.8** | **43.3** |

Table 5: Comparisons for language-guided object detection on PASCAL VOC 2007. Models under transductive setting are trained on COCO dataset.

| Method | Setting | Image Encoder | Text Encoder | mAP |
|---|---|---|---|---|
| Faster R-CNN Ren et al. (2015) | Inductive | VGG | - | 73.2 |
| Faster R-CNN Ren et al. (2015) | Inductive | RN101 | - | 75.2 |
| YOLO v2 Redmon & Farhadi (2017) | Inductive | Darknet-19 | - | 78.6 |
| CenterNet Zhou et al. (2019) | Inductive | DLA-34 | - | **80.7** |
| DenseCLIP Rao et al. (2022) | Transductive | CLIP-ViT-B | CLIP-B | 74.1 |
| UniBoost (Ours) | Transductive | BEiT-B | T5-B | **77.4** |

objects. We evaluate our UniBoost for language-guided object detection and instance segmentation on COCO dataset under supervised setting shown in Table 4 and transductive setting Table 5.

**Experiment Setup.** During inference, the maximum side of input image is set to 800 for model with transformer-based backbones while others adopt $1333 \times 800$. We train the model for a total of 12 epochs with a batch size of 16. We use the AdamW optimizer and a stepwise learning rate decay scheduler with a base learning rate of $2e^{-4}$, which decays in epoch 8 and 11 with a ratio of 0.1.

**Results.** Table 4 evaluates UniBoost on COCO dataset. Compared to the model with image-text pair pre-trained weights, our UniBoost with unsupervised pre-trained unimodal BEiT-B and T5 outperforms by 1.7% (42.6%→44.3%) and 1.2% (39.6%→40.8%) box AP and mask AP under supervised setting, respectively, especially ours adopts a smaller input size.

Additionally, we conduct transductive experiments to evaluate the generalization performance of our model on object detection. In our transductive experiment, the models are trained on the COCO dataset, which contains a large number of object detection images with diverse object categories and backgrounds. We then evaluate the model on the the PASCAL VOC 2007, which contains a different set of object categories and backgrounds compared to COCO. By evaluating the models on a different dataset than the one used for training, we can gain insights into how well the models can generalize to new, unseen examples. This is particularly important in object detection, where the ability to detect objects accurately in a wide range of scenarios is critical for real-world applications. As shown in Table 5, our UniBoost achieves larger performance promotion by 3.3% (74.1%→77.4%) compared to the CLIP-based model and are even comparable with some supervised models.

Table 6: Comparisons for visual question answering on VQA v2.0 benchmark.

| Method | # Parameters | Test-dev |
|---|---|---|
| VLKD | 832M | 44.5 |
| Flamingo3B Alayrac et al. (2022) | 3.2B | 49.2 |
| BLIP-2 Li et al. (2023) | 3.1B | 49.7 |
| UniBoost | 935M | 58.4 |

## 4.3 VISUAL QUESTION ANSWERING

VQA, as the most typical vision-language task, requiring the model to simultaneously understand the context of images and texts, and thus can be used to evaluate the capability of a vision-language

model. VQA can be solved by treating it as a close-set multi-choice problem or as open-set answering generation problem. Here, we consider VQA task as an answer generation problem following Li et al. (2023), and evaluate UniBoost on VQA v2.0 Antol et al. (2015) under zero-shot setting. Note that we do not perform any specific fine-tuning after multitask fine-tuning. The results are shown in Table 6.

**Results.** As shown, our UniBoost outperforms Flamingo3B by 9.2% and BLIP-2 by 8.7% with much fewer parameters, which attributes to unsupervised unimodal pre-training and multitask fine-tuning.

Table 7: Comparisons for image captioning on NoCaps validation set.

| Method | # Parameters | CIDEr | SPICE |
|---|---|---|---|
| VinVL Zhang et al. (2021a) | 345M | 105.1 | 14.4 |
| BLIP Li et al. (2022b) | 446M | 113.2 | 14.8 |
| Flamingo Alayrac et al. (2022) | 10.6B | - | - |
| SimVLM Wang et al. (2022b) | 1.4B | 112.2 | - |
| BLIP-2 Li et al. (2023) | 1.1B | 121.0 | 15.3 |
| CoCa Yu et al. (2022) | 2.1B | **122.4** | **15.5** |
| UniBoost | 935M | 119.1 | 15.1 |

## 4.4 IMAGE CAPTIONING

Image captioning is another task which requires the model to embody a comprehensive understanding of language and image content. After multitask fine-tuning, following Yu et al. (2022) we fine-tune UniBoost without network modification only on COCO captioning task without CIDEr optimization and evaluate on the validation set of NoCaps Agrawal et al. (2019). During fine-tuning, we train UniBoost for 10K steps with a warm-up of 1000 steps. The batch size and learning rate is set to 64 and 1e-5 respectively.

**Results.** As shown in Table 7, our UniBoost achieves comparable performance with CoCa but with much fewer trainable parameters, and also achieve comparable performance with BLIP-2 but with much less training data, i.e. BLIP-2 requires pre-training on 170M image-text pairs while ours are trained on 6.5M multitask data.

## 5 LIMITATION

While our current framework and experiments focus on image and text data, it is important to note that the latest multimodal models, such as ImageBind Girdhar et al. (2023), include audio and other modalities in their embedding space. By incorporating other modalities, these models are able to capture a wider range of contextual information, leading to more different types of applications. Although our proposed framework has shown promising results in enhancing the zero-shot performance of vision-language tasks, it is important to continue studying the applicability of our method to other modalities. We believe that the inclusion of other modalities can further improve the quality and diversity of pre-training. This can potentially lead to even better performance in downstream tasks by enabling the model to better understand the complex relationships between different modalities via our UniBoost framework.

## 6 CONCLUSION

In this paper, we validate that unsupervised unimodal pre-training can significantly boost the performance of zero-shot vision-language tasks, in comparison against supervised pre-training or image-text pair pre-training. In principle, unsupervised pre-training can make use of not only image-text pair data or their intersection, but also image and text data on their own, the union of which cover a much broader and more diverse data distribution compared to image-text pairs. To take this advantage, we introduce UniBoost, a multitask fine-tuning framework based on unsupervised unimodal encoders. UniBoost can both leverage the generalization power of unsupervised unimodal embedding and learn a broader joint image-text solution space by incorporating multitask supervised data, thereby improving zero-shot performance on downstream tasks.

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
