# UniBoost: Boost Zero-shot Vision-Language Tasks via Multitask Fine-tuning with Unsupervised Unimodal pre-training

## A   Appendix

### A.1   Ablation Study

We provide the performance contribution analysis through conducting ablation studies on unsupervised unimodal pre-training and multitask fine-tuning. Unless stated, otherwise the multitask fine-tuning settings are consistent with the ones detailed in the main paper.

**Effect of Unsupervised Unimodal pre-training.** Our UniBoost aims to benefit from the unsupervised pre-trained unimodal models which are not restricted to the intersection of image and text data. We replace the supervised encoders by class labels or language with unsupervised encoders step-wisely from the first row to the last row in Table 1 and observe continuous performance promotion.

**Effect of multitask Fine-tuning.** Another source for the performance promotion is the usage of large-scale supervised data across tasks. We compare the models with and without multitask fine-tuning as shown in the 4th-5th rows of Table 1. Zero-shot performance of language-guided semantic segmentation on PASCAL-$5^0$ is gained by 2.8% (64.1→67.3). Another notable observation is that CLIP encoders benefit less (61.9→62.5) from multitask fine-tuning compared to unsupervised pre-trained unimodal encoders (64.1→67.3).

**Effect of Different Tasks.** The goal for our multitask fine-tuning is to learn general and robust multimodal alignment or fusion from supervised data instead of achieving state-of-the-art performance with specific tuned parameters on each tasks. In this ablation study, we explore the effect of individual dataset on different tasks, and train each model for 500K steps during multitask fine-tuning while other settings are unchanged. The results are evaluated on PASCAL-$5^0$, validation set of COCO detection dataset, VQA v2.0 dataset and NoCaps dataset. As shown, these tasks especially the zero-shot tasks can be benefit from a joint training of multiple tasks.

Table 1: Ablation study on different pre-trained image and text encoders.

| Image Encoder | Text-Encoder | Multitask Fine-tuning | Pascal-$5^0$ |
|---|---|:---:|---|
| ViT-L | CLIP-B | | 61.3 |
| CLIP-L | CLIP-L | | 61.9 |
| CLIP-L | CLIP-L | ✓ | 62.5 |
| MAE-L | CLIP-B | | 64.1 |
| MAE-L | CLIP-B | ✓ | 67.3 |
| MAE-L | T5-S | ✓ | 68.7 |

### A.2   Implementation Details

**Network of UniBoost-Neck.** We illustrate the network structure of the neck in Figure 1. In vision tasks, to aggregate multi-level information, we apply feature pyramid network following the conventional recognition methods, in which $3 \times 3$ conv-bn-relu layers are applied and $C_i$ is set to 512 or 768 in base or large model respectively. In vision-language alignment tasks, we employ a separate projection layer on the image feature from FPN and text feature respectively to align the embedding space dimension (512 is adopted). Afterwards, we compute the similarity score through dot production between the two served as classification results. In vision-language fusion tasks, we apply a multimodal transformer followed by a language head.

Table 2: Ablation study on different tasks.

| Model | PASCAL-$5^0$ | COCO Detection | VQA v2.0 | NoCaps |
|---|---|---|---|---|
| UniBoost | 66.5 | 43.7 | 54.7 | 117.1 |
| w/o ImageNet | 65.6 | 42.9 | 53.8 | 116.3 |
| w/o PASCAL | 63.3 | 43.5 | 54.5 | 117.3 |
| w/o COCO Detection | 64.2 | 42.0 | 53.5 | 116.7 |
| w/o VQA v2.0 | 66.2 | 43.4 | 48.3 | 115.5 |
| w/o NoCaps | 66.0 | 43.1 | 50.9 | 113.3 |

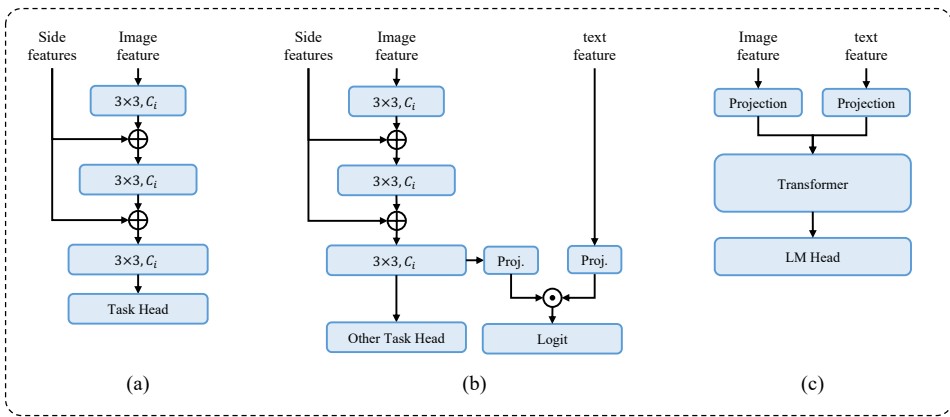

Figure 1: Network structure of the UniBoost-Neck.

**Datasets and Multitasking.** Our UniBoost is fine-tuned on multiple tasks as summarized in Table 3 to learn the alignment or fusion between visual and textural embedding space. During training, we group all the data by batch and iterate batches in each round. To balance the weights of different tasks during training, especially for those with a small number of total training images, e.g., less than 64,000 images, we augment the images corresponding to these tasks multiple times by random crop and random resizing with scales ranging from 0.8 to 1.2, so that the number of total training images is around 64,000 at least which is an empirical value.

### A.3 MORE EXPERIMENTAL RESULTS

**Language-guided Semantic Segmentation.** We give the zero-shot results of language-guided semantic segmentation evaluated on PASCAL-$5^i$ and COCO-$20^i$. To test the robustness and generalization ability of zero-shot semantic segmentation models across different domains of datasets, we evaluate the models trained on different split of PASCAL-$5^i$ on COCO dataset. Since PASCAL VOC and COCO data share some classes, we remove all the common classes during evaluation to make sure all the evaluated classes are novel to the model. Table 5 tabulates the results, where we provide the performance of LSeg with CLIP-RN101 for reference since the weights of LSeg with ViT-L are not available. As shown, our models trained on different splits of PASCAL-$5^i$ achieves higher and more stable performance on COCO dataset as LSeg exhibits a large gap among the models trained on different splits. Additionally, we provide qualitative results in multiple scenarios including indoor, landscape and street views on ADE20K dataset in Figure 2–4. As shown, our method boosts the classification or pixel grouping accuracy, thus improves the segmentation performance benefiting from more general and robust representation.

**VQA.** We compare our method with other multitask methods on VQA v2.0 benchmark in Table 5. We finetune our UniBoost on VQA v2.0 dataset by treating VQA as a multi-choice classification task. As shown, ours outperforms Unified-IO Lu et al. (2023) by a large margin and is also superior to X-Decoder Zou et al. (2023).

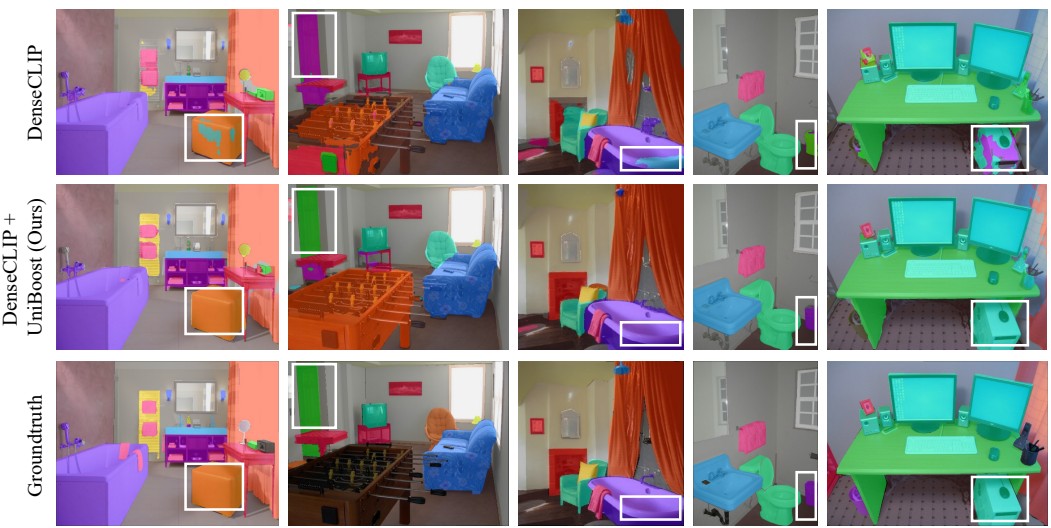

Figure 2: Qualitative results on indoor samples in ADE20K.

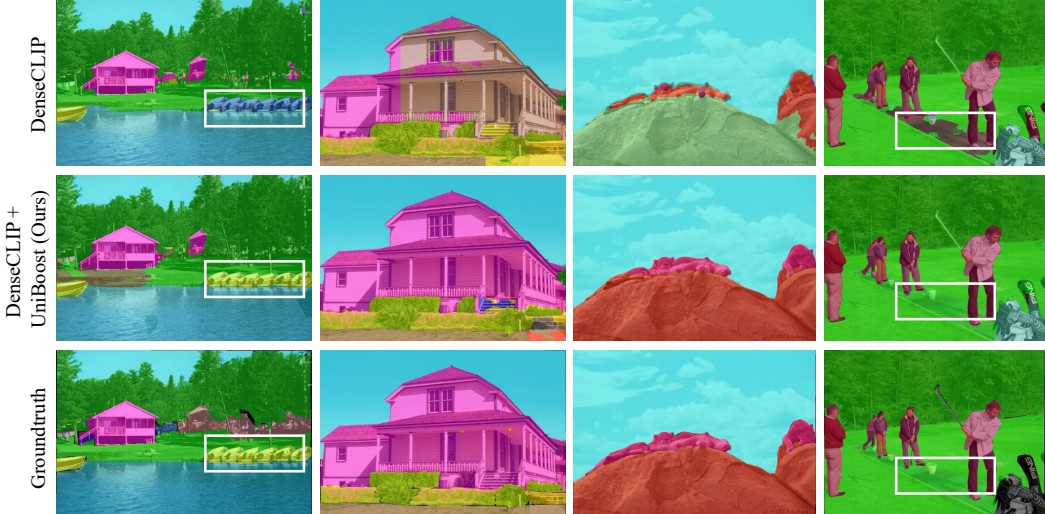

Figure 3: Qualitative results on landscape samples in ADE20K.

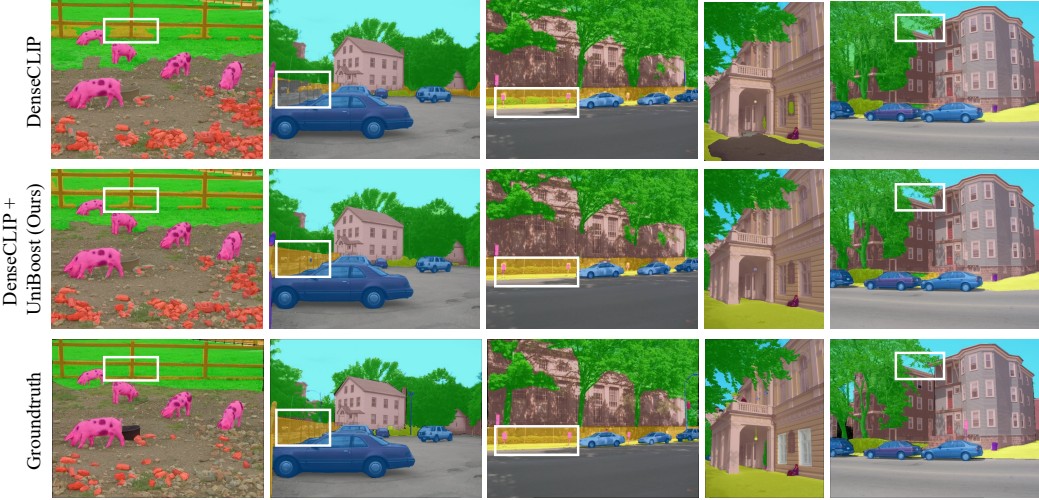

Figure 4: Qualitative results on street views in ADE20K.

Table 3: Tasks and datasets used for multitask fine-tuning.

| Task | Dataset | Number of Images |
|---|---|---|
| Image classification | ImageNet-1K Deng et al. (2009) | 1,000,000 |
| Object detection | COCO Lin et al. (2014)
Objec365 Shao et al. (2019) | 118,287
2,000,000 |
| Instance segmentation | COCO Lin et al. (2014) | 118,287 |
| Semantic segmentation | ADE20K Zhou et al. (2017)
PASCAL Context Mottaghi et al. (2014) | 20,000
10,103 |
| Language-guided detection | COCO Lin et al. (2014) | 118,287 |
| Language-guided segmentation | ADE20K Zhou et al. (2017)
PASCAL Context Mottaghi et al. (2014)
FSS1000 Li et al. (2020) | 20,000
10,103
10,000 |
| Depth estimation | NYUv2 Nathan Silberman & Fergus (2012) | 1,449 |
| Denoising | SIDD Abdelhamed et al. (2018) | 30,000 |
| Deblurring | GoPro Nah et al. (2017) | 3214 |
| Image captioning | COCO Lin et al. (2014)
Nocaps Agrawal et al. (2019)
SBU captions Ordonez et al. (2011)
Conceptual Captions Sharma et al. (2018) | 82,783
15,100
860,000
1,000,000 |
| Visual questioning answering | VQA v2.0 Antol et al. (2015)
Visual Genome Krishna et al. (2017) | 82,783
101,174 |
| Visual Reasoning | NLVR2 Suhr et al. (2019) | 107,292 |

Table 4: Evaluation on COCO of language-guided semantic segmentation models trained on different splits of PASCAL-$5^i$. Since the weights of LSeg with CLIP-ViT-L are not available, we provide the performance of LSeg with CLIP-RN101 for reference.

| Method | Image Encoder | Text Encoder | PASCAL-$5^i \rightarrow$ COCO | | | | |
|---|---|---|---|---|---|---|---|
| | | | $5^0$ | $5^1$ | $5^2$ | $5^3$ | mean |
| LSeg | CLIP-RN101 | CLIP-B | 24.93 | 23.88 | 17.03 | 27.04 | 23.22 |
| UniBoost | MAE-B | T5-S | **26.66** | **28.04** | **27.10** | **28.60** | **27.60** |

Table 5: Evaluation results of multitask methods on VQA v2.0.

| Method | Test-dev | Test-std |
|---|---|---|
| Unified-IO Lu et al. (2023) | 71.6 | - |
| X-Decoder Zou et al. (2023) | 76.8 | 77.0 |
| UniBoost | **77.8** | **77.9** |