# OpenReview forum: "UniBoost: Boost Zero-shot Vision-Language Tasks via Multitask Fine-tuning with Unsupervised Unimodal Pre-training"
_ICLR.cc/2024/Conference — Submitted to ICLR 2024_

### Official Review · Reviewer_tHnr · 2023-10-21

**Soundness:** 3 good
**Presentation:** 3 good
**Contribution:** 3 good
**Rating:** 5
**Confidence:** 3

**Summary:**

The main idea of the paper is to use unsupervised uni-modal pre-trained modality encoders (vision and text) in a new framework to learn how to solve tasks. Usually, previous works use vision-text pre-trained modality encoders which limits the ability of these pre-trained models as the datasets that contain vision-text pairs may not be that big and they may also contain a lot of noise. Thus, the paper proposes the use of unsupervised uni-modal pre-trained modality encoders which can take advantage of uni-modal data and as they do not require any supervision, they can be pre-trained on massive datasets. Moreover, the paper introduces UniBoost, a method to take advantage of these unsupervised pre-trained uni-modal models such that they can be combined to do even multi-modal tasks.

**Strengths:**

- The paper is clear and easy to follow, containing only minor typos.
- There are comparisons with recent baselines.
- The proposed method is tested on different datasets and tasks, showing that it can achieve state-of-the-art performance in all of them.

**Weaknesses:**

- Right below Fig.3, there is a “.” alone on the line. Remove it.
- Again, right below Fig.3 “and architecture modification Zhou et al. (2021). multitask fine-tuning (MTF) extends”. I think “multitask” should start with a capital letter.
- “it is found that TS-MTL often underperforms single ask trained models;”. Is “ask” here referring to “task”?
- “The neck works bridges the embeddings and the task head”. Did you want to say “the neck works by bridging the…”? Or “The neck bridges…”?
- Fig 4 for me is a bit unclear. Looking at it without reading the text I would assume that there are 3 very independent systems that are not linked at all. However, reading the text it seems that they are linked. The “Conv” from (a) is also used in (b) for the image modality, right? If so, I would say to make this explicit in the figure. Moreover, E_i and E_t from equation (1) are the same as the ones from the section “Image-text alignment tasks”? If so, then also make this explicit in the figure. Mainly, I think it would be good to show the relation between all these 3 systems so that one can look at the figure and easily get a high-level idea of the paper without any confusion.
- Table 1. Regarding the difference in performance between the LSeg and the first UniBoost (the one that uses a different image encoder). The text says that this difference in performance shows that unsupervised pre-trained encoders are better than the supervised ones. However, UniBoost also uses a new method for training (by using the UniBoost-Neck and multi-task fine-tuning). Thus, in this experiment two elements are changed compared to LSeg: the encoders but also how they are trained. I would say that this experiment does not necessarily show this. I would like to see the experiment with UniBoost and with the supervised encoder. If that shows that supervised encoders obtain lower performance, then the claim is correct in my view. This claim is also made in Table 2, Table 3, in the introduction and also in the conclusion.

**Questions:**

- Most of my comments can be addressed easily as they are mostly about the paper presentation and typos.
- However, the paper claims multiple times that “we validate that unsupervised unimodal pre-training can significantly boost the performance of zero-shot vision-language tasks, in comparison against supervised pre-training or image-text pair pre-training”. This is also the first sentence in the conclusion. I am not sure about this claim. As mentioned in the weakness section, when these experiments are done, the encoder but also the way of training the whole system changes, by employing the UniBoost-Neck. Thus, it can be that the improvements are due to the UniBoost-Neck and not due to the encoders. I would like to see the experiments where the authors employ supervised pre-trained models in UniBoost and show how the performance changes.

---

### Official Review · Reviewer_eQGs · 2023-10-31

**Soundness:** 2 fair
**Presentation:** 1 poor
**Contribution:** 2 fair
**Rating:** 3
**Confidence:** 3

**Summary:**

The paper introduces UniBoost, a method that leverages unsupervised pre-trained unimodal models to boost the zero-shot performance in vision-language tasks. The primary contribution is a multitask fine-tuning framework that is built upon separate unsupervised pre-trained vision and language encoders. This design enables the model to capitalize on the advantages of both unsupervised pre-training and a diversified set of supervised aligned data.

**Strengths:**

- The paper addresses a potentially impactful research topic within the domain of vision-language tasks.
- A commendable breadth of experiments was conducted, even though there are significant concerns to address, as detailed in the weaknesses section.

**Weaknesses:**

## Presentation
- The manuscript presentation and clarity could benefit from substantial improvements. For instance, sentences such as, "However, pre-training with image-text pairs limits itself to cover a wide range of unimodal data, where noise can also be introduced as misaligned pairs during pre-processing" (first lines of the abstract) are challenging to decipher.

## Novelty
- The reviewer acknowledge to not be deeply familiar in the topic, thus possibly missing context and details. However, a potential oversight in the paper is the neglect of the research direction of WVLP. Specifically, quoting from [1] : "[...] explore how to utilize lowcost unimodal data with limited cross-modal supervision, i.e., weakly supervised vision-and-language pre-training (WVLP).", seems pertinent and overlapping to the presented work.

- The authors should consider [1] as a comparative baseline. Indeed, [1] reports much higher results than the ones presented in the current work, e.g. compare Table 6 of the manuscript to Table 2 in [1].

---

I would like to emphasize that being hard to follow compromise the overall value of the work. Such lack of clarity impedes comprehension of the paper core contributions and does not align with the standards expected for ICLR.

---

[1] Chen, C., P. Li, M. Sun, and Y. Liu (July 2023). “Weakly Supervised Vision-and-
Language Pre-training with Relative Representations”. In: Proceedings of the 61st
Annual Meeting of the Association for Computational Linguistics. ACL 2023. Toronto, URL: https://aclanthology.org/2023.acl-long.464

**Questions:**

- In the manuscript it is claimed that "Recently, intermediate fine-tuning has been inserted before fine-tuning the pre-trained models on the target task for preventing overfitting and enhancing adaptation". However, this claim lacks citations. Providing pertinent references would be beneficial, especially for readers who are unfamiliar  in this domain.

---

### Official Review · Reviewer_BiMn · 2023-11-01

**Soundness:** 3 good
**Presentation:** 2 fair
**Contribution:** 1 poor
**Rating:** 3
**Confidence:** 4

**Summary:**

- The paper presents UniBoost, an approach for vision-language modeling wherein pretrained SSL models trained on unimodal data are used as initialization for the image and text encoders before multitask finetuning the fused model on more limited image-text data
- Uniboost shows SOTA results for COCO-20i and PASCAL-5i language-guided semantic segmentation and gets strong results on language-guided object detection, instance segmentation, VQA, and image captioning

**Strengths:**

- UniBoost has strong results on multiple evaluations (although some important details are missing, see Weaknesses)
- The captioning results on NoCaps and VQA results on VQAv2 are quite strong given that UniBoost is smaller and sees fewer image-text pairs

**Weaknesses:**

- Uniboost is described as "a multitask fine-tuning framework based on the unsupervised pre-trained vision and language models", but this is not a new contribution. Significant works in the space leverage unimodally pretrained  multimodal models, e.g. FLAVA intializes from a pretrained DINO model and a language encoder trained on CCNews and BookCorpus. BLIP-2 leverages pretrained models as well, but keeps them frozen. Regarding the part about using "unsupervised" pretrained encoders, I fail to see why unsupervised is added as a qualifier -- there are no ablations showing why an unsupervised MAE / BEiT encoder is better than a supervised encoder (e.g. CLIP).
- There are no ablations in the paper, it is just a set of comparisons against various other methods, where Uniboost does comes out on top (except for an ablation in Tables 1, 2 where the text encoder is varied and T5-S is shown to outperform CLIP-B). It is unclear whether the gains are because of the improved pretrained encoders, different training dataset, or any other differences. There are no details mentioned in the paper about the training data used. Table 1 and 2 are called zero-shot, but does this mean COCO is not used during pretraining? The UniBoost model used in the tables also changes (Tables 1 and 2 use MAE-L init, whereas Tables 3, 4, 5 use BEiT-B), there should be a single model evaluated on all the tasks. The model parameters are also not mentioned, except for Tables 6 and 7 where it is not clear what the architecture used is
- The experimental comparisons are also system level ones and it isn't clear why only certain comparisons were made. For Table, 3 which is a fully supervised setting, MAE ViT-L achieves 53.6 mIoU but the paper doesn't compare to this and a lot of other works. In section 4.2 the evaluation is called language-guided but not zero-shot, why is that the case? Why are the comparisons only made against DenseCLIP? Works like GLIP [1] get much stronger results zero-shot.
- Minor: In Table 6 "BLIP-2" is misleading since the best BLIP-2 result is 65.0, the table should clarify this.

[1] Li, Liunian Harold, et al. "Grounded language-image pre-training." Proceedings of the IEEE/CVF Conference on Computer Vision and Pattern Recognition. 2022.

**Questions:**

- What are the key contributions from the paper? If it is initializing from unsupervised encoders how is this different from prior work? Why are unsupervised encoders important? It would be important to have ablations to showcase the importance of the key contributions, which is something that is missing from the paper
- There are a lot of missing details in the paper (see weaknesses) around the data used, the model parameters, the comparisons, whether the setup is zero-shot or not, and why only certain works are compared in the tables. Why does the model architecture and initialization change depending on the evaluation?

---

### Meta-Review · Area_Chair_c5Lr · 2023-12-11

**Metareview:**

This paper presents a new framework named UniBoost. It takes the late fusion approach that combines unimodal encoder representations in a late fusion stage, thus each unimodal encoder can be pre-trained using modality specific self-supervised learning, together with a cross-modal training objective. And the authors demonstrate the multitask fine-tuning on top of the pre-trained models can achieve strong performance on downstream tasks.


While the work exhibits clear motivation and achieves strong results, its novelty is somewhat constrained. The utilization of pre-trained unimodal encoders is a prevalent practice in this field, and it has been employed in numerous prior works, as indicated by reviewers BiMn and eQGs. Furthermore, the reviewers note that the experimental results may not entirely substantiate the authors' assertion regarding the effectiveness of unsupervised unimodal pre-training.

**Justification For Why Not Higher Score:**

It is a clear rejection given the reviewers' comments.

**Justification For Why Not Lower Score:**

N/A

---

### Decision · Program_Chairs · 2024-01-16

Reject